# The Oligosaccharide Region of LPS Governs Predation of *E. coli* by the Bacterivorous Protist, *Acanthamoeba castellanii*

Ying Liu,[a] Gerald B. Koudelka[a]

[a]Department of Biological Sciences, University at Buffalo, Buffalo, New York, USA

**ABSTRACT** Protozoan predation is a major cause of bacterial mortality. The first step of predation for phagocytic amoebae is the recognition of their prey. Lipopolysaccharide (LPS) is a major component of Gram-negative bacteria and is only present on the outer leaflet of the outer membrane lipid bilayer. LPS consists of three distinct regions: lipid A, an oligosaccharide core, and *O*-polysaccharide. Previous research in our lab determined that the oligosaccharide (OS) region of LPS mediates the recognition and internalization of *Escherichia coli* by *Acanthamoeba castellanii*. The oligosaccharide region is conceptually divided into the inner core and outer core. The LPS of any given *E. coli* strain contains only one of five different OS structures: K-12 and R1 to R4. All OSs contain the same inner core sugars but different outer core sugars. Here, we show that the Kdo2 moiety of the inner core is necessary and sufficient for *E. coli* recognition and internalization by *A. castellanii*. We also show that the precise composition of the variable outer core OS region modulates the efficiency with which *A. castellanii* consumes bacteria. The latter finding indicates that outer core OS composition plays a role in bacterial defense against phagocytic predators.

**IMPORTANCE** Rather than being transmitted from host to host, most opportunistic bacterial pathogens reside in the environment for significant amounts of time. Protist predation is a major cause of bacterial mortality. To enhance their survival in the environment, bacteria have evolved various defense strategies such as filamentation, increased motility, biofilm formation, toxin release, and modification of cell wall structure; strategies which also enhance their virulence to humans. This work shows that the major component of the bacterial cell wall, LPS, also known as bacterial endotoxin, is a "dual use" factor, regulating amoeba predation of bacteria in addition to its well-known role as a human virulence factor. Both these functions are governed by the same parts of LPS. Thus, the structure and composition of this "dual use" factor likely evolved as a response to constant voracious protist grazing pressure in the environment, rather than during short-term infections of human and animals.

**KEYWORDS** amoebae, bacterivores, cell wall, Gram-negative bacteria, lipopolysaccharide, predation, protists

Predation is a major cause of bacterial mortality (1). Predation of bacteria by protists effectively controls the structure and activity of microbial communities in many ecosystems (2–4). Successful predation of bacteria by protists consists of three distinct stages: recognition, internalization, and digestion. If prey can block any of these steps of predation, predation fails. Therefore, to survive, bacteria have evolved many different defense methods to resist protist predation (1, 5). These strategies include modifying the surface of the cell.

Bacterial lipopolysaccharides (LPS) comprise the major outer surface membrane components of almost all Gram-negative bacteria. Therefore, the bacterial surface, hence, LPS, is the first line of defense against antimicrobial molecules and stress caused by changes in the environment surrounding the bacterium. These molecules are also the

Address correspondence to Gerald B. Koudelka, koudelka@buffalo.edu.

The authors declare no conflict of interest.

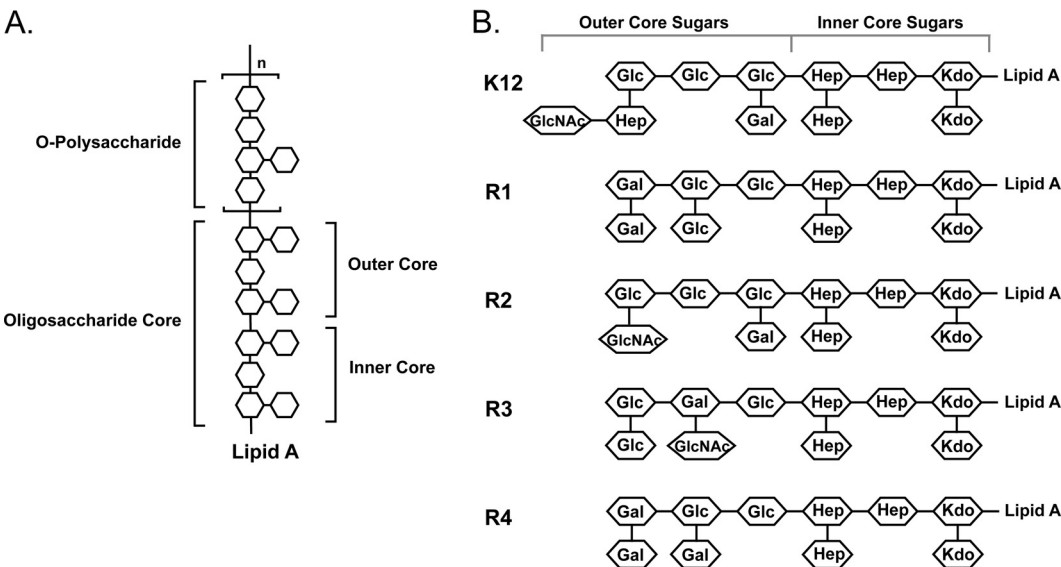

**FIG 1** Structures of lipopolysaccharide (LPS). (A) Shown is a simplified diagram of LPS structure. LPS consists of three moiety: lipid A, which anchors the LPS onto the outer membrane; an oligosaccharide (OS) core; and repeating *O*-antigen polysaccharide. (B) Structure and composition of the five different OS core types: R1, R2, R3, R4, and K-12 expressed by *Escherichia coli*. The identities of the component sugars are indicated inside the hexagons (Gal, D-galactose; GalNAc, *N*-acetyl-D-galactosamine; Glc, D-glucose; GlcNAc, *N*-acetyl-D-glucosamine; Hep, heptose; Kdo, 3-deoxy-D-manno-oct-2-ulosonic acid). The OS can be conceptually divided into two regions: the inner core which is proximal to lipid A, and the outer core sugars. The inner core sugars are highly conserved while the outer core sugars display more diversity.

first ones encountered by other elements of the environment. As such, these extracellular polysaccharides play a variety of important roles in the life of bacteria, e.g., enabling bacterial evasion of the host immune system and mediating interactions with other cells and bacteriophages. We and others have found that the bacterial predation efficiency of protists is affected by the identity of the carbohydrates present on the bacterial cell surface (6–8).

LPS consists of a hydrophobic lipid A, and two carbohydrate components: (i) an oligosaccharide (OS) core, and (ii) *O*-antigen (9, 10) (Fig. 1A). Lipid A is a glucosamine-based phospholipid that functions as the hydrophobic anchor to secure the LPS molecule in the outer membrane of Gram-negative bacteria. The OS core consists of a short chain of sugars which can be further divided into two regions: the inner core which is proximal to lipid A, and the outer core which provides the attachment site for *O*-antigen.

*O*-antigens consist of repeating subunits of three to five sugars (11). The repeating unit can be linear or branched. *O*-antigen composition is highly diverse, containing >60 different monosaccharides and 30 different noncarbohydrate components (10). Individual *O*-antigen chains vary in length as well. The structure and composition of *O*-antigens of *Escherichia coli* and several other Gram-negative bacterial species are especially variable, with more than 186 *O*-antigen forms having been identified in *E. coli*. Similarly, the enteric pathogen *Salmonella enterica* presents at least 46 different *O*-antigens. In contrast, the OS region of LPS in Gram-negative bacteria displays more limited diversity. The structure of the inner core region of OS is nearly invariable and is primarily composed of L-glycero-D-manno-heptose (Hep) and 3-deoxy-D-manno-oct-2-ulosonic acid (Kdo) residues. The structure and composition of the outer core sugars in the OS region are more variable, with *E. coli* expressing five different OS types (R1, R2, R3, R4, and K-12), (Fig. 1B) while only two different OS types are found in *Salmonella*. Regardless of the diversity of *O*-antigens and OS, each different bacterial strain presents only one OS type and either one or no *O*-antigen type.

*Acanthamoeba castellanii*, a free-living protist, is a main consumer of bacteria in many ecosystems. *In vitro* studies have shown that predation of *E. coli* by *A. castellanii* is modulated, in part, by the type of *O*-antigen present on the bacterial surface (8).

Several lines of evidence indicate that the interaction of the carbohydrate portion of LPS with the membrane-bound mannose-binding protein of *A. castellanii* governs the efficiency of bacterial consumption by *A. castellanii* (8). Consistent with these findings, intestinal protozoa recognize antigenically diverse *Salmonella* with different efficiencies (6). In this case, prey discrimination is seemingly solely determined by the identity of the *O*-antigen (6). However, Arnold et al. (8) found that *O*-antigen is not required for recognition of *E. coli* by *Acanthamoeba*. Instead, their findings suggest that carbohydrates in the OS region of the LPS are the element used by *Acanthamoeba* to detect *E. coli*. Given the indication that *Acanthamoeba* recognizes its bacterial prey using the OS region of LPS, we explored the role of this region in governing the efficiency of protist predation.

The backbone of the core OS consists of 3-deoxy-D-manno-oct-2-ulosonic acid and heptose and hexose sugars. The genes encoding the glycosyltransferases for core sugars assembly are located at the *waa* locus (12). Defined by the first gene in each transcriptional unit, three operons have been found in the *waa* locus: *gmhD*, *waaQ*, and *waaA* (10). The *waaA* operon contains the structural genes for the Kdo addition of the inner core. The product of *gmhD* transcription is responsible for the biosynthesis and transfer of L,D-heptose. The heptosyltransferases encoded in this operon, *waaC* and *waaF*, incorporate L,D-heptose into the inner core. The central *waaQ* operon encodes genes that are necessary for the biosynthesis of the outer core.

In this paper, we examined the role of outer core sugars in regulating bacterial uptake by *Acanthamoeba*. We determined the ability of amoebae to consume five different *O*-antigen-deficient *E. coli* strains, each of which contained one OS type: R1, R2, R3, R4, or K-12. We found that uptake efficiency varies significantly with OS type. We also examined the efficiencies with which *A. castellanii* consumed a series of *waa* mutant K-12 and R3 *E. coli* strains. These strains each bear a mutation in different glycosyltransferase genes and thereby contain LPS that is truncated at a precise sugar in the OS region. We found that truncating a particular OS at specific sugars alters the efficiency with which *A. castellanii* consume these strains. This observation shows that the precise OS composition regulates bacterial recognition by amoebae. Nevertheless, all these strains are recognized and internalized by *A. castellanii*. Consistent with this result, we also found that K-12 Δ*waaC*, which contains the two Kdo of inner core sugars, can be recognized and internalized by *A. castellanii*. Additionally, treating amoeba with Kdo2-lipid A, but not lipid A alone, inhibits bacterial internalization. This result indicates that the two lipid A-attached Kdo inner core sugars are both sufficient and necessary for bacterial uptake by *Acanthamoeba*.

## RESULTS

**Structure and sequence of the LPS oligosaccharide region regulates bacterial recognition by *Acanthamoeba*.** Our previous findings indicate that *Acanthamoeba* detects *E. coli* by recognizing the carbohydrates in the OS region of LPS (8). To identify the precise determinants in the OS region which allow the amoeba to recognize *E. coli*, we examined the ability of *Acanthamoeba* to consume five different *O*-antigen-deficient *E. coli* strains, each of which bore one of the five different extant OS types: R1, R2, R3, R4, and K-12 (Fig. 1B). As a negative control, we also included ATCC 11775, an *E. coli* strain which we have previously shown is not consumed by *A. castellanii* (8).

For these experiments, the five different *E. coli* OS strains were separately seeded onto proteose peptone glucose (PPG) agar plates. Subsequently, amoebae were spotted at the center of each plate. If the amoebae consume the bacteria, they form a plaque in the bacterial lawn. The size of this plaque is proportional to the efficiency with which the amoebae consume a given bacterial strain. Amoebae were incubated on these plates for 5 days at 30°C prior to measurement of plaque size.

Acanthamoebae formed plaques on all the *O*-antigen-deficient OS variant *E. coli* strains (Fig. 2). This indicates that all of these strains, R1, R2, R3, R4, and K-12, are recognized and consumed by acanthamoebae. This observation confirms our earlier finding that *Acanthamoeba* recognizes bacteria via the OS region (13). However, although

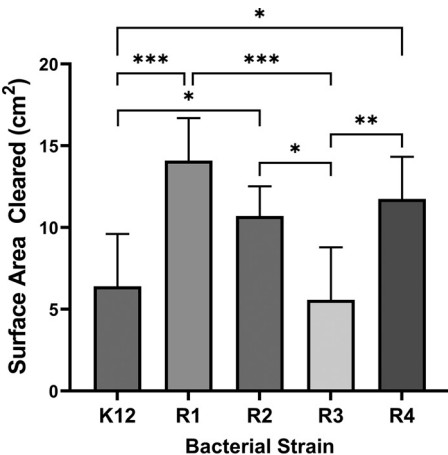

**FIG 2** Effect of OS type on *Acanthamoeba* predation of *E. coli*. *Acanthamoeba* ($10^6$ cells in a 10-$\mu$L volume) were separately spotted on bacterial lawns of 5 different *E. coli* strains, each bearing one of the 5 different LPS OS types, K-12 (MG1655), R1, R2, R3, and R4, growing on proteose peptone glucose (PPG) plates. Amoebae were allowed to grow on these plates for 5 days at 28.5°C. The sizes of plaques formed on bacterial lawns by grazing amoeba were determined using ImageJ software. Error bars represent standard deviations from ≥3 independent quadruplicate measurements. Significance was determined by a one-way analysis of variance (ANOVA) followed by *post hoc* implementation of Tukey's multiple-comparisons correction. Comparisons were made using the adjusted *P* value to determine the statistical significance of pairwise comparisons. ***, $P < 0.005$; **, $P < 0.01$; *, $P < 0.05$; NS, not significant ($P > 0.05$).

acanthamoebae consumed all these strains, the relative size of the amoeba plaque, and thereby the efficiency with which the amoebae consumed each strain, varied significantly with OS type. For example, the size of the amoebae plaque formed on the *E. coli* strain bearing the R1 OS was >2 times larger than that formed on R3. Similarly, the size of the amoeba plaque formed on R2 was also larger than that on R3. Since the carbohydrate composition and sequence of the inner core region of the OS was identical in all R-type strains, but the outer core varied (Fig. 1B), this suggests that the composition of the OS outer core region regulates the efficiency of bacterial recognition and consumption by *Acanthamoeba*.

The OS outer core region in all R-types consists of a tri-hexose backbone (HexI-HexII-HexIII), in which HexI, the first sugar, is always glucose, but the residues of HexII and Hex III and the sugars attached to each of the three backbone hexoses vary according to OS type. Of the five different OS types, R1 and R4 are the most similar in structure, differing from each other only by the identity of the sugar residue which branches off from the glucose that occupies the HexII (second) position in these R-types (Fig. 1B). In R1, the branching residue is glucose, while in R4 this residue is galactose. Consistent with their structural similarity, there was no significant difference in the efficiency with which amoebae consumed R1 and R4 (Fig. 2).

K-12 and R2 OS outer core regions have an identical tri-hexose backbone, consisting of three glucose residues with a Gal residue branching off the glucose at HexI. However, the structure and composition of the outer core regions of these OS regions differ with respect to the sugar substituent on the glucose at HexIII. K-12 bears a disaccharide containing $\beta$-D-GlcNAc-(1,7)-L-$\alpha$-D-heptose attached to this glucose. Similar to the other R-type outer cores, the hexose in the third position in R2 is attached to monosaccharide substituent; here, $\alpha$-1,2-*N*-acetylglucosamine occupies this position. Acanthamoebae consumed K-12 less efficiently than they did R2, suggesting that the presence and/or sequence of a disaccharide attached to the third glucose may impede amoeba recognition of K-12 strains (Fig. 2).

The composition of the outer core backbone of R3 is unique among the outer core regions, consisting of glucose at HexI, galactose at HexII, and glucose at Hex III (Fig. 1B). In addition, the R3 outer core is the only one in which the HexIII residue has a Glucose substituent. Also, the second galactose contains $\alpha$-1,3-*N*-acetylglucosamine as a substituent.

An $\alpha$-1,3-*N*-acetylglucosamine residue is not found in the outer core OS region of R1 and R4 but is only found in the K-12 OS. The amoebae plaque formed on the R3 strain was significantly smaller than those formed on the R1 and R4 strains (Fig. 2). The amoebae plaque formed on the R3 strain was essentially identical in size to that formed on the K-12 strain (Fig. 2).

**Specific determinants in the outer core OS region regulate bacterial recognition by amoebae.** Together, the results shown in Fig. 2 show that the structure and/or composition of the outer core region of the OS modulates amoebae recognition of their bacterial prey. More precisely, these findings indicate that amoebae avoid consuming bacteria containing disaccharide and/or an *N*-acetylglucosamine substituent(s) in this region. To begin to distinguish between these alternatives and identify which sugars in the outer core affect bacterial consumption by amoebae, we examined the ability of Acanthamoebae to consume a series of K-12 and R3 *waa* mutant strains in which the OS region was truncated at a precise outer core sugar (Fig. 3). For these experiments in K-12, in-frame single-gene deletions of *waa* glycosyltransferases (Fig. 3A) were obtained from the Keio collection (14). For R3 (Fig. 3B), an in-frame, tagged-deletion $\Delta waaD$ mutant strain was created using the suicide recombinant vector pKOV (15). The in-frame disruption of the R3 glycosyltransferase-encoding genes, *waaI*, *waaJ*, and *waaG* was constructed by insertion of a chloramphenicol-resistance cassette using phage $\lambda$ Red recombinase (16) (see Materials and Methods). As expected, the *waa* deletions in R3 sequentially reduced the molecular weight of the LPS (Fig. S1), indicating that deletions of these genes resulted in the predicted change in structure.

In all OS types, the first hexose of the trisaccharide backbone (HexI-HexII-HexIII) of the outer core is always a glucose residue (Fig. 3A and B). This residue is inserted by the WaaG glucosyltransferase (17, 18). Because this Glc-(1,3)-Hep linkage defines the junction between the inner and outer core regions, knocking out *waaG* in K-12 or R3 results in LPS containing only the conserved inner core sugars. We found that *waaG* deletion greatly enhanced amoeba consumption of K-12 and R3 strains (Fig. 3C and D, respectively). This result is consistent with the suggestion that the sequence and/or composition of OS outer core region downregulates amoeba recognition of these *E. coli* strains (Fig. 2, see Discussion) This finding also suggests that the determinant used by amoebae to recognize prey is located in the OS inner core.

In K-12, the HexIII glucose is added by the WaaR enzyme. Therefore, $\Delta waaR$ strains contain only the HexI-HexII of the outer core backbone (Fig. 3A). We found no significant difference in the ability of amoebae to consume the K-12 strain, which bears the complete OS region, and K-12 $\Delta waaR$, which lacks the HexIII and the Hep-GlcNAc side chain linked to it (Fig. 3C). This observation indicates that in K-12, the sugars at HexIII and/or those attached to it do not affect the ability of Acanthamoebae to recognize its bacterial prey.

HexII in K-12 is added by the WaaO enzyme (18, 19), meaning that K-12 strains with a WaaO deletion contain an LPS that only contains HexI in the outer core of the OS region (Fig. 3A). We found that Acanthamoebae consumed K-12 $\Delta waaO$ more efficiently than K-12 $\Delta waaR$ (Fig. 3C). This finding suggests the HexII sugar negatively impacts the ability of amoeba to recognize K-12 *E. coli*. This suggestion is consistent with the observation that Acanthamoebae consumed the K-12 $\Delta waaO$ strain more efficiently than MG1655, a strain that bears the complete K-12 outer core OS region. Together, these results indicate that the presence of glucose at HexII regulates the ability of *Acanthamoeba* to recognize K-12 bacterial strains.

In R3 strains, the glucose at HexIII is added by the glucosyltransferase encoded by *waaJ* (Fig. 3B). This residue also bears a glucose substituent, which is added by the WaaD glucosyltransferase. The amoeba consumption efficiency of R3 strains bearing the *waaD* deletion was identical to that of the R3 prototype (Fig. 3D). This result suggests that the substituent of HexIII on R3 does not influence amoeba recognition of bacteria. The LPS of the R3 $\Delta waaJ$ strain lacks both the terminal HexIII glucose residue and its substituent, but contains the HexI-HexII of the outer core backbone and the GlcNAc substituent residue linked to HexII. The R3 $\Delta waaJ$ strain was consumed at an efficiency equal to that of the unmodified R3 strain, indicating that HexIII does not influence amoeba recognition.

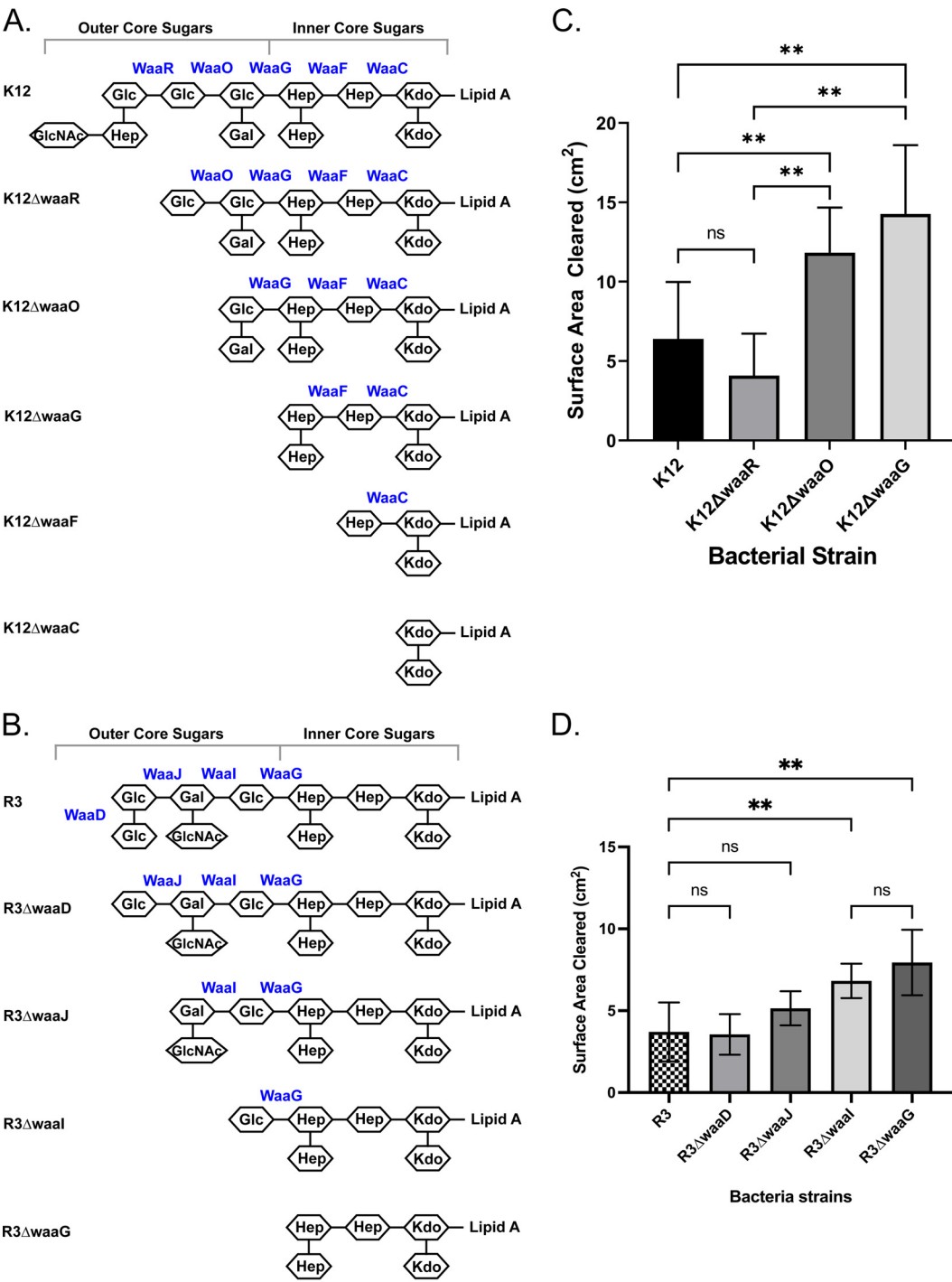

**FIG 3** Role of OS outer core residues on *Acanthamoeba* consumption of *E. coli*. (A and B) Structures of the OS regions of LPS of K-12 (A) and R3 (B), respectively. The identities of the sugars are indicated inside the hexagons (see Fig. 1 legend) and the genes encoding the glycosyltransferase enzymes known to add each sugar are indicated next to the vertical lines. (C and D) Effects of mutating these enzymes on *Acanthamoeba* predation of *E. coli* bearing the K-12 and R3 OS core types, respectively. For these experiments, *E. coli* strains MG1655 (K-12) and R3, bearing deletions of the indicated glycosyltransferase-encoding genes, were separately seeded on PPG plates, and then *A. castellanii* ($10^6$ cells in a 10-$\mu$L volume) was spotted on the center of the plate. Amoebae were allowed to grow on these plates for 5 days at 28.5°C. The sizes of the plaques formed on bacterial lawns by grazing amoeba were determined and analyzed as described in Materials and Methods and the Fig. 2 legend.

The glycosyltransferase responsible for the adding the galactose residue at HexII into the outer core of R3 OS is encoded by *waaI* (17), meaning that the R3 Δ*waaI* mutation eliminates addition of both the HexII galactose and its GlcNAc substituent, leaving only the glucose at HexI in the outer core OS of the R3 LPS (Fig. 3B). We found that Acanthamoebae consumed R3 Δ*waaI* more efficiently than the prototype R3 strain, suggesting that HexII galactose and/or its GlcNAc substituent residue play a role in regulating amoeba recognition of this strain (Fig. 3D).

**Inner core Kdo residues govern bacterial recognition by amoeba.** The results shown in Fig. 3 demonstrate that the specific sugars located in the outer core of the OS region of LPS regulate Acanthamoebae consumption of bacteria. However, our results with the *waaG* deletion strains also indicate that that the determinant required by amoebae to recognize and consume their prey is located in the OS inner core (Fig. 3). We wished to identify this determinant. To do this, we examined the ability of Acanthamoebae to consume several bacterial strains, each of which bears a deletion of one the genes which encodes the glycosyltransferases responsible for assembling the OS inner core region. Because the inner core structure and composition is identical for all *E. coli*, for these experiments, we used the Keio strains which bear deletions of these genes in the K-12 background.

The WaaC and WaaF heptosyltransferases sequentially assemble the heptose residues in the inner core (Fig. 3A). Deletion of the *waaF* gene prevents addition of the diheptose residue at the outer edge of the inner core region, whereas deletion of the *waaC* gene results in an LPS that contains only the two Kdo residues attached to lipid A (Fig. 3A). We found that amoeba consumption of the K-12 Δ*waaF* and K-12 Δ*waaC* mutant strains was identical, and significantly higher than the consumption of a K-12 strain containing the complete OS region (Fig. 4). These findings suggest that the two Kdo residues of the conserved inner core sugars are sufficient for the recognition and uptake of *E. coli* by *Acanthamoeba*.

The two Kdo residues attached to lipid A are needed to maintain the integrity of Gram-negative cell wall under our growth conditions. We have previously shown (8) that adding purified LPS or mannose can block bacterial recognition by competing with bacteria for binding to the *Acanthamoeba* mannose-binding protein (aMBP) which is located on the surface of the amoeba. Therefore, to explore the importance of Kdo in bacterial recognition by amoebae, we examined whether purified lipid A and/or Kdo2-lipid A could compete with the uptake of green fluorescent protein (GFP)-labeled MG1655 by *Acanthamoeba*. For this assay, amoebae were first separately incubated with or without lipid A or Kdo2-lipid A for 1 h at room temperature, then cocultured with MG1655 expressing GFP for 1 h at room temperature, followed by 2 h at 30°C. The effect of lipid A and Kdo2-lipid A on bacterial recognition and uptake was measured by determining the fraction of amoebae that contained GFP-labeled bacteria. In the absence of any additions, we found that ~28% of the amoebae contained GFP-labeled bacteria under these conditions, which is the maximum amount (100%) of amoebae that consumed GFP-labeled bacteria under our conditions. Incubating amoebae with 40 μg/mL Kdo2-lipid A prior to the addition of GFP-labeled *E. coli* reduced the fraction of GFP-containing amoebae by ~49%, whereas adding an identical amount of lipid A did not reduce uptake of GFP-labeled *E. coli* by amoebae (Fig. 5). We obtained similar qualitative and quantitative results using a gentamicin protection assay (Fig. S2) (7). These findings, together with our previous research (8), indicate that the two Kdo residues in the inner core sugars of *E. coli* LPS are necessary and sufficient for the bacterial recognition and consumption of *Acanthamoeba*.

Previous studies indicate that *Acanthamoeba* recognizes bacterial prey via its mannose binding protein (20). To further explore this recognition mechanism, we determined whether mannose or a Kdo monosaccharide not attached to a lipid A moiety was able to compete with bacterial uptake by amoebae. The presence of 3 mM mannose reduced bacterial uptake by ~55% (Fig. 6), which is consistent with our previous observations (8). However, the addition of up to 3 mM Kdo monosaccharide did not significantly affect bacterial uptake by amoebae (Fig. 6). The latter finding suggests that amoebae use the Kdo2 moiety as the bacterial recognition determinant.

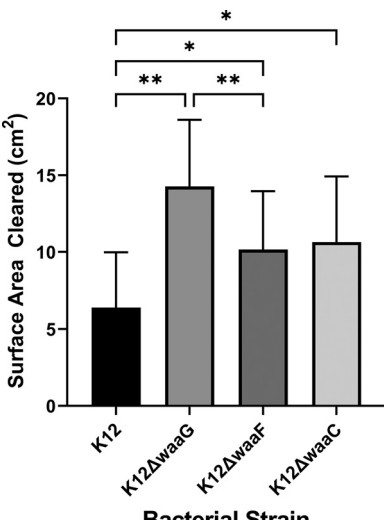

**FIG 4** Role of OS inner core residues on *Acanthamoeba* consumption of *E. coli*. *E. coli* strain MG1655 (K-12), bearing deletions (14) of the indicated glycosyltransferase-encoding genes responsible for adding OS inner core residues, was separately seeded on PPG plates, and then *A. castellanii* ($10^6$ cells in a 10-$\mu$L volume) was spotted on the center of the plate. Amoebae were allowed to grow on these plates for 5 days at 28.5°C. The sizes of the plaques formed on bacterial lawns by grazing amoeba were determined and analyzed as described in Materials and Methods and the Fig. 2 legend Significance was determined by a one-way ANOVA followed by a *post hoc* implementation of Tukey's multiple-comparisons correction. Comparisons were made using the adjusted *P* value to determine the statistical significance of pairwise comparisons. **, $P < 0.01$; *, $P < 0.05$; NS, not significant ($P > 0.05$).

## DISCUSSION

*Acanthamoeba* internalizes bacteria using receptor mediated phagocytosis (21). Previous research in our lab has shown that the recognition and uptake of *E. coli* by *A. castellanii* is governed by the sugar residues of the *E. coli* LPS OS region (8). Each *E. coli* presents one of five different OS types, R1, R2, R3, R4, and K-12, on its cell surface. The

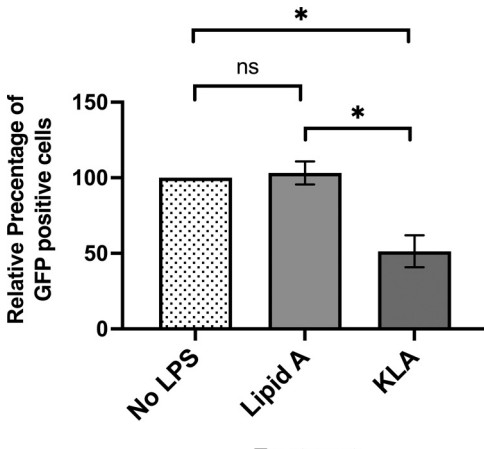

**FIG 5** Effect of added lipid A variants on *Acanthamoeba* consumption of *E. coli*. Effect of incubating *A. castellanii* without or with either lipid A or Kdo2-lipid (KLA) on consumption of MG1655 expressing green fluorescent protein (GFP). Amoeba were first separately incubated with or without the indicated amount and type of lipid A and then cocultured with MG1655 expressing GFP as described in Materials and Methods. Cocultures were subsequently processed as described in Materials and Methods. The fraction of GFP-positive amoebae was measured using a LUNA-FL Dual Fluorescence Cell Counter. The percentage of GFP-positive amoebae relative to that seen in the absence of added LPS is shown. Error bars represent standard deviations of ≥3 independent experiments, each with at least two technical replicates. Significance was determined using a one-way analysis of variance followed by a *post hoc* implementation of Tukey's multiple-comparisons correction. Comparisons were made using the adjusted *P* value to determine the statistical significance of pairwise comparisons. **, $P < 0.01$; *, $P < 0.05$; NS, not significant ($P > 0.05$).

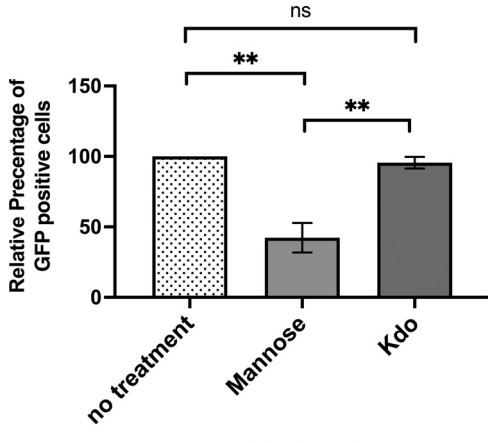

**FIG 6** Effect of added monosaccharides on *Acanthamoeba* consumption of *E. coli*. Effect of incubating *A. castellanii* with or without either mannose or Kdo on consumption of MG1655 expressing GFP. Amoebae were first separately incubated with or without the indicated monosaccharide and then cocultured with MG1655 expressing GFP as described in Materials and Methods. Cocultures were subsequently processed as described in Materials and Methods. The fraction of GFP-positive amoebae was measured using a LUNA-FL Dual Fluorescence Cell Counter. Percentage of GFP-positive amoebae relative to that seen in the absence of added LPS is shown. Error bars represent standard deviations from ≥3 independent experiments, each with at least two technical replicates. Significance was determined using a one-way ANOVA followed by a *post hoc* implementation of Tukey's multiple-comparisons correction. Comparisons were made using the adjusted $P$ value to determine the statistical significance of pairwise comparisons. **, $P < 0.01$; *, $P < 0.05$; NS, not significant.

OS region consists of a variable outer core segment and a completely conserved inner core region (Fig. 2A). Here, we show that *Acanthamoeba* recognizes and consumes *E. coli* bearing the various R types (Fig. 2B), albeit with varying efficiencies. The finding that all *E. coli* can be recognized and internalized by amoebae, irrespective of R-type, indicates that the determinant recognized by this organism lies in the conserved inner core OS region. The variability in uptake of *E coli* containing different R-types shows that the sugars in the variable outer core OS region regulate the ability of amoebae to recognize the determinant located in the inner core OS region.

We acknowledge that it is possible that the effect of changing OS type and/or removing various OS carbohydrates could affect *Acanthamoeba* consumption of bacteria by differentially altering its ability to digest the various strains and/or impacting the nutritional quality of the bacterial prey. However, given that the changes in OS framework are relatively minor, and that LPS form a minor component of the nutrients in bacteria, it is difficult to imagine that these features would significantly affect these attributes of the prey. Consequently, this observation, together with our other evidence, strongly supports the idea that OS affects amoeba consumption of bacteria by directly affecting the efficiency with which amoebae recognize their prey.

The inner core OS is comprised of Kdo and L-glycero-D-manno-heptose (L,D-Hep) residues (Fig. 1B). Three lines of evidence indicate that Kdo is the determinant in the inner core OS that is recognized by *Acanthamoeba*: (i) the K-12 Δ*waaC* mutant *E. coli*, whose LPS contains only the two Kdo residues, is recognized and consumed by *Acanthamoeba* (Fig. 4); (ii) purified Kdo2-lipid A inhibits the uptake of MG1655 by Acanthamoebae (Fig. 5, Fig. S2); and (iii) purified lipid A does not inhibit the uptake of MG1655 by Acanthamoebae (Fig. 5, Fig. S2). Taken together, these results show that the lipid A-attached Kdo2 moiety of the conserved LPS inner core is necessary and sufficient for recognition and consumption of *E. coli* by *A. castellanii*.

Recognition and uptake of *E. coli* by *Acanthamoeba* is mediated by its mannose-binding protein, aMBP, which is located on its surface (8). There are no mannose residues in the *E. coli* OS region. Our finding that Kdo is the determinant recognized by *A. castellanii* indicates that aMBP mediates recognition of *E. coli* in a manner similar to other mannose-binding

lectins; that is, aMBP can bind a diverse set of sugars (22). Consistent with this idea, as with *E. coli* (8), the phagocytosis of two other Gram-negative bacteria, *Legionella pneumophila* (23) and *Arcobacter butzleri* (24), whose LPS OS also contains Kdo, is also blocked by exogenous mannose. A role for aMBP in the recognition and uptake of *L. pneumophila* by *A. castellanii* has also been established. Thus, *A. castellanii* leverages Kdo, a ubiquitous and highly conserved feature of the Gram-negative bacteria cell wall, and aMBP to recognize this group of food sources.

At first consideration, the close juxtaposition of Kdo with lipid A and the bacterial outer cell wall might be expected to inhibit the ability of aMBP to access the Kdo disaccharide. However, aMBP is not unique in its ability to bind Kdo-lipid A; several other proteins that recognize bacteria are known to do so via this sugar. For example, Kdo attached to lipid A is known to be an important factor in enhancing the innate immune responses mediated by the Toll-like receptor 4/MD2 (25), and this residue is directly contacted by amino acids in this protein (26). Similarly, other phagocytic cells also target the core sugars for recognition. The social amoeba *Dictyostelium discoideum* uses a G-protein-coupled receptor, folic acid receptor 1, to recognize the core sugars of LPS and facilitate engulfment of Gram-negative bacteria (27). Macrophages also recognize the core sugars of LPS and promote the engulfment of the Gram-negative enteric pathogen *Salmonella enterica* serovar Typhimurium using its receptor brain-specific angiogenesis inhibitor 1, which belongs to the adhesion-type family of 7-transmembrane receptors (28, 29). Moreover, there is evidence showing that inner core sugars interact with cell surface proteins. The innate immune protein human surfactant protein D, a collagenous C-type lectin, has been found to specifically target the highly conserved LPS proximal inner core Hep-Kdo motif from *S. enterica* and *H. influenzae* (22, 30). The mannose-binding lectin (MBL), which is a human serum protein and originally described as the murine Ra-reactive factor, interacts with the heptose of inner core from *Yersinia enterocolitica* (31). Together with our results (Fig. 4), these findings show that cell surface receptors can interact with the innermost sugars of the OS region and mediate engulfment of bacteria.

Despite convincing evidence that *A. castellanii* recognizes *E. coli* by binding to Kdo, we found that exogenously added Kdo monosaccharide does not inhibit bacterial uptake by *Acanthamoeba* (Fig. 6). It is unclear why this is the case. We envision two possibilities. First, it is possible that the Kdo disaccharide is required for high-affinity binding to aMBP. This possibility is consistent with the finding that monomeric Kdo-lipid A less efficiently activates cytokine production by elements of the innate immune system than does Kdo2-lipid A (25). Alternatively, the conformational flexibility of unliganded Kdo monosaccharide may prevent its recognition, and the structural limitations presented by anchoring the Kdo monosaccharide to the lipid bilayer of membrane may be required for its efficient recognition by *Acanthamoeba*.

Previous studies in our lab showed that some types of *O*-antigen block bacterial recognition and internalization by *A. castellanii* (8), that is, O1 *O*-antigen-containing LPS does not interact with aMBP, while O157 *O*-antigen-containing LPS does. However, which specific characteristic of this carbohydrate chain (length, structure, composition) contributes to the blockage of recognition by *Acanthamoeba* remains unknown. Nevertheless, this finding shows that carbohydrates can interfere with bacterial recognition by *A. castellanii* and provide resistance to phagocytosis.

Compared to *O*-antigen, the variety, structure and composition of the *E. coli* outer core region is much less complex. *O*-antigen is a polysaccharide composed of multiple oligosaccharide repeating units, each with two to seven residues, whereas the outer core of each OS type contains a trisaccharide backbone with two or three branches. The composition of outer core is relatively simple and consists of common monosaccharides such as glucose, galactose, or *N*-acetylglucosamine (32). Regardless of this simplicity, we find that *A. castellanii* differentially consumes *E. coli* containing different R-types. This finding shows that the variable carbohydrate composition of the outer core OS region

regulates the ability of amoebae to recognize the determinant located in the inner core OS region. In particular, we found that of all R-types, R3 and K-12 were consumed least efficiently, suggesting that the sugar composition of these R-type OS regions is most effective at blocking this recognition (Fig. 2). Consistent with this idea, we found that removing the outer core sugars by deleting *waaG* in both K-12 and R3, which resulted in strains that only contain inner core OS on their LPS, increases amoeba consumption of these strains. Our results indicate that the HexII glucose in K-12 and the HexII galactose and/or its GlcNAc substituent residue in R3 are the primary determinants that govern the ability of *Acanthamoeba* to discriminate between *E. coli* strains containing those two OS types (Fig. 3C and D). This phenomenon is similar to the case of *Klebsiella pneumoniae*, where the outermost glucose of core sugars of its LPS modulate resistance to phagocytosis by the amoeba *D. discoideum* (33).

The mechanism by which these sugars decrease *Acanthamoeba* consumption of bacteria is unclear. They may block aMBP from accessing the Kdo2 moiety or they may interfere with phagocytosis. Regardless, our results show that the addition of one or two common sugars in the LPS outer core increases a bacterium's resistance to recognition and consumption by its predators. Our observations suggest that the LPS-associated carbohydrate of Gram-negative bacterial LPS outer may form part of an antipredator defense strategy against phagocytosis.

## MATERIALS AND METHODS

**Strains, plasmids, and chemicals.** *A. castellanii* was a gift from Wendy Trzyna, Marshall University. *E. coli* K-12 strain MG1655 was purchased from the Coli Genetic Stock Center (https://cgsc.biology.yale.edu/). *E. coli* strain ATCC 11775 was purchased from the American Type Culture Collection (Manassas, VA). The *E. coli* prototype R strains R1(F470), R2 (F632), R3 (F653), and R4 (F2513) were gifts from Chris Whitfield (University of Guelph, Canada). The *E. coli* K-12 single-gene deletion mutant strains K-12 Δ*waaR*, K-12 Δ*waaO*, K-12 Δ*waaG*, K-12 Δ*waaF*, and K-12 Δ*waaC* were from the Keio collection (14). pKOV was obtained from Addgene. pKD3 was obtained from the Coli Genetic Stock Center. pKD46-RecX was a gift from Michael Berger, University of Muenster. A plasmid encoding GFP was obtained from James Hurst, Washington State University. Kdo2-lipid A and lipid A were purchased from Cayman Chemical Company (Ann Arbor, MI). Kdo monosaccharide (3-deoxy-D-manno-2-octulosonic acid-ammonium salt) was purchased from Toronto Research Chemicals (Toronto, Ontario Canada).

**Cultivation and harvesting of *A. castellanii*.** *A. castellanii* were grown in 5 mL ATCC medium: 712 PYG with additives at 30°C in flat-bottomed 25-mL tissue culture flasks. After 4 days of stationary incubation, growth media were decanted from flasks and 5 mL fresh medium was added to each flask. Flasks were placed on ice for 20 min to displace amoeba from the bottom of the flask and cells were then collected for passaging or experimentation. Stocks of *A. castellanii* were made and stored at 4°C as described previously (34). *Acanthamoeba* were harvested for predation assay via centrifugation at 300 × $g$ at room temperature, washed three times with Page's amoeba saline (PAS), and resuspended in PAS.

**Chromosomal gene inactivation in R3.** R3 Δ*waaD* was created using suicide recombinant vector pKOV as described (15). The following primers were used to amplify DNA fragments upstream and downstream of *waaD*: *waaD*-Ni: 5′-cccatccactaaacttaaacatattattttatcaaccatctcgtatattaaccttc-3′; *waaD*-No: 5′-cgcggatcctctggtgtaatgtatattaatttacgggaatgg-3′; *waaD*-Co: 5′-cgcggatcccaaactaatagttcttatcgctatgcaaatgg-3′; and *waaD*-Ci: 5′-tgtttaagtttagtggatgggcaaagattcgaagaggttattcataattggtttg-3′.

R3 Δ*waaJ*, R3 Δ*waaI*, and R3 Δ*waaG* were created using phage λ Red recombinase as described previously (14). All N-terminal deletion primers bore an 80-nt 5′ extension including the gene initiation codon, and all C-terminal deletion primers consisted of 21 nt for the C-terminal region, the termination codon, and 59 nt downstream. The following primers were used to amplify DNA fragments of the chloramphenicol-resistance gene from pKD3: R3-*waaJ*-up: 5′-agaaaattaaaaaagtaatcagggatgtaaaagtcaaactag gattgaagagcaaataatgattatgatgaagggtaattatgggaattagccatggtcc-3′; R3-*waaJ*-down: 5′-aggaggaattgaaaaa ataggagtaaccgtaaatattatttttatcaaccatctcgtatattaacctttcataacattatagtgtaggctggagctgcttc-3′; R3-*waaI*-up: 5′-aa atcagggaaagaacgattcgaaaatcgttgtaacccgtaacttattttttgccaaaattttggatacagaataaatatgatgggaattagccatggtcc-3′; R3-*waaI*-down: 5′-ctttcaacccatttatcgttttattatatatcataatgaattattttaaccttaaatctttagaagcattttctttatagtgtaggctg gagctgcttc-3′; R3-*waaG*-up: 5′-ctttcggttattccggcggcagatgtcattgctgctgtcgataaaattactgccctcctccacgacaggtacg tcgttatgatgggaattagccatggtcc-3′; and R3-*waaG*-down: 5′-ttttaacttcctcaaaaggatctttgccgcgccataacgtggcaa acggctctttaagttcaaccatccagaccacccgtgtgtaggctggagctgcttc-3′. R3 (F653) carrying the Red helper plasmid pKD46-RecX was grown in 30 mL of LB supplemented with 100 $\mu$g/mL ampicillin and 100 mM L-arabinose at 30°C to an optical density at 600 nm (OD$_{600}$) of 0.6. Cells were washed three times with ice-cold 10% glycerol and resuspended in sterile double-distilled water. Next, 50 $\mu$L of cells was mixed with 600 ng of the PCR fragment in an ice-cold 0.2-cm cuvette. Cells were electroporated at 2.5 kV with 25 mF and 200Ω, immediately followed by the addition of 1 mL of SOC medium (2% bacto tryptone, 0.5% yeast extract, 10 mM NaCl, 2.5 mM KCl, 10 mM MgCl$_2$, 10 mM MgSO$_4$, 20 mM glucose) with 1 mM L-arabinose. After overnight incubation at 37°C, the mixture was spread onto an agar plate with 15 $\mu$g/mL chloramphenicol to select CmR transformants at 37°C. Verification of correct chromosomal insertion was carried out by PCR with chloramphenicol gene-specific primers and locus-specific primers. The following primers were used for

verification: *waaJ*-up-VF: 5′-atgcgactgaatccaaccct-3′ and *waaJ*-down-VR: 5′-gatatgggatagtgttcacccttgtta-3′; *waaI*-up-VF: 5′-ttacgtgccgatcaacgtct-3′ and *waaI*-up-VR: 5′-cgtcgttggcgggataaaga-3′; *waaG*-down-VF: 5′-agtat-caacgccgacatccc-3′ and *waaG*-down-VR: 5′-cgacgatttccggcagtttc-3′.

**Purification and visualization of LPS.** Purification of LPS from R3 and its *waa* mutants was performed as described (16). Briefly, bacterial cells were washed and suspended in SDS buffer (2% β-mercaptoethanol, 2% SDS, and 10% glycerol in 0.1 M Tris-HCl [pH 6.8]) and boiled for 15 min. The samples were treated with 0.25 mg/mL DNase I, 0.25 mg/mL RNase A, and 0.5 mg/mL proteinase K. The LPS molecules were purified by hot phenol extraction. LPS purity was assessed by electrophoresis 14% SDS-PAGE followed by silver-staining.

**Predation assay.** A predation assay was performed essentially as described previously (8). Briefly, proteose peptone glucose agar plates were separately seeded with 100 μL of saturated cultures (~$10^9$ cells/mL) of different bacterial strains and incubated for 1 h at 37°C. Subsequently, $10^5$ *A. castellanii* trophozoites in 10 μL of Page's Amoeba Salts were spotted on each plate. Each plate was sealed with parafilm and incubated at 28.5°C for 5 days. After incubation, the surface area cleared by the amoebae on each plate was measured and recorded. Each measurement was done in quadruplicate and averaged. The data shown represent ≥3 independent quadruplicate measurements.

**Competition assay with lipid A and Kdo2-lipid A.** For the competition assay, 400 μL of freshly harvested 4- to 6-day-old amoebae was aliquoted into each well of a 24-well polystyrene tissue culture plate and incubated at 30°C overnight. After removing the planktonic cells suspended in medium, the amoeba cells that had adhered to the bottom of each well were washed at room temperature three times with PAS (35, 36). *E. coli* MG1655 expressing GFP were grown in LB supplemented with 30 μg/mL kanamycin overnight. Bacteria were washed at room temperature three times with PAS. Amoeba were incubated with 300 μL PAS and 40 or 80 μg/mL Kdo2-lipid A at room temperature for 1 h and then cocultured with $6 \times 10^8$ *E. coli* MG1655 expressing GFP for 1 h at room temperature, followed by 2 h of incubation at 30°C. After incubation, the amoebae were washed three times with PAS and the plates were placed on ice for 20 min to release amoeba cells for measurement. The fraction of GFP-positive amoebae was measured using a LUNA-FL Dual Fluorescence Cell Counter (Logos Biosystems, Annandale, VA).

## SUPPLEMENTAL MATERIAL

Supplemental material is available online only.
**SUPPLEMENTAL FILE 1**, PDF file, 0.2 MB.

## ACKNOWLEDGMENT

This research received no specific grant from any funding agency in the public, commercial, or not-for-profit sectors.

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
