## [Reviewer comments · Microbiology Spectrum]

Microbiology Spectrum

The oligosaccharide region of LPS governs predation of *E. coli* by the bacterivorous protist, *Acanthamoeba castellanii*

Ying Liu and Gerald Koudelka

Corresponding Author(s): Gerald Koudelka, University at Buffalo

Review Timeline:

Submission Date:	September 13, 2022
Editorial Decision:	November 2, 2022
Revision Received:	December 12, 2022
Accepted:	December 21, 2022

Editor: Olaya Rendueles

Reviewer(s): The reviewers have opted to remain anonymous.

Transaction Report:

DOI: <https://doi.org/10.1128/spectrum.02930-22>

November 2, 2022

Dr. Gerald B Koudelka
University at Buffalo
Biological Sciences
611 Cooke Hall, North Campus
Buffalo, NY 14260

Re: Spectrum02930-22 (The oligosaccharide region of LPS governs predation of *E. coli* by the bacterivorous protist, *Acanthamoeba castellanii*)

Dear Dr. Gerald B Koudelka:

Thank you for submitting your manuscript to Microbiology Spectrum.

I am very sorry for the delay in your manuscript.

Unfortunately, I had a lot of difficulties securing reviewers. I have just received the last review this morning.

As you can see, two reviewers have read your manuscripts, and they both find it interesting but raise some concerns, specifically in respect to the statistical analyses. Please make sure that these analyses are carried out appropriately and that your data support your conclusions.

Link Not Available

Sincerely,

Olaya Rendueles Garcia

Journals Department
Reviewer comments:

Reviewer #1 (Comments for the Author):

Comments to authors

In their manuscript, the authors addressed the role of specific LPS moieties in the predation of *E. coli* by the amoebae *A. castellanii*. In a co-culture assay, they determined the predation efficiency by measuring the size of plaques caused by the predator in an *E. coli* lawn, *E. coli* strains with specific LPS properties were systematically analyzed. Furthermore, purified LPS moieties were tested for their ability to compete with *E. coli* uptake when they were directly added to amoebae. The authors conclude that the Kdo disaccharide is necessary and sufficient for prey recognition and that prey uptake is further regulated by sugar residues in the outer core oligosaccharide of LPS.

The work addresses the very interesting molecular mechanisms of microbial predation, many of which are unknown. The paper is clearly written and the argumentation of the authors can be followed. Nevertheless, I have a few concerns and comments that I would please ask the authors to address.

Figure 1A and B.

This Figure is very important to follow the paper, but it is also difficult to read due to its small labeling. Please amend. Moreover, I suggest to include labels for Kdo and HexI-II to 1A, since these terms are used repeatedly.

Fig. 2 and all similar graphs:

In what unit is the relative area measured, on the cm or mm scale? I would find this information helpful to interpret the data.

Fig. 3A/B: Again, these Figures are not readable for small labels, and 3A also for poor resolution. Please amend. Furthermore, it appears that in 3B the labels below (description of mutants) were accidentally overlaid?

Fig. 3C: The labeling of statistical significance is unclear. What significance is supposed to be shown by the bar second from top, which is drawn between *waaR*, *waaO* and *waaG* mutants? Please clarify.

I. 237/ Fig. 3D: Is the difference between R3 and R3*waaI* statistically significant? If yes, please include the respective label to the figure. If no, please clearly state in the text.

Fig. 3D and corresponding text I. 203-206: I agree with the statement that deletion of *waaR* enhances consumption of K12 and R3. However, I do not understand why in this assay consumption of K12 and R3 are so different from each other, when in Fig. 2 the authors point out that they are almost identical? Please clarify.

I. 254 and throughout: Please be consistent the designation of mutant strains with or without colon.

I. 261: It is not clear to me how this experiment would work - do the authors expect that adding purified LPS components would block a receptor, e.g. aMBP? If so, this should be explained more clearly.

Figure 5 is identical with Figure 6, presumably a mistake in image upload. Please present the correct figure for the data described in the text for Fig. 5.

Reviewer #2 (Comments for the Author):

In this paper, Koudelka et al. is described how specifically the oligosaccharide region of LPS dictates the sensitivity of *E. coli* to predation by the protist *Acanthamoeba castellanii*. Using multiple strains, authors demonstrate that the strains having similar OS structures are similar at predation sensitivity. Further, the authors describe how the successive changes in the outer sugar moieties can influence the sensitivity of *E. coli* to predation. In addition, they also showed how the Kdo-Kdo moiety of the inner core plays a vital role in the recognition of the prey *E. coli* by the predator *Amoeba*. However, I have a few concerns regarding the analysis of the data and its interpretation.

1. It is not clear if the authors have corrected for multiple testing in Figure - 2. To the naked eye, it looks as if the multiple testing will reveal that difference is not significant in most cases. Moreover, it is not clear why did the authors use MG 1655 as a reference for their analysis? Especially when ATCC11775 can be used as a control, which is in fact even more resistant to predation than the MG strain.

2. Similarly data from the MG1655 strain is used to normalize the data in figure 3. Why not use respective parental strain as a reference? For instance, when the deletion mutants of R3 were described, MG1655 was taken as a control to normalize the data. Why not use R3 instead to normalise the data?

3. A minor comment about the Figures 3 C and D: "Bacterial strains" is mentioned twice. Authors can remove the heading from this figure. This will allow them to show the results from the statistical test clearly. For example, it is not clear whether the difference between *dwaaR*, *dwaag*, and *dwaaO* in figure 2C is significant or not.

4. Line 206 to 208 is not clear. Seems like a part of the sentence is missing. More importantly, it is not clear to me that the

differences seen here are because of differences in the ability of predators in recognizing the prey. I suspect that such differences may arise because of differential ability to digest the prey, differential nutritional quality, or differences in the ability to move on the respective prey types. The information provided in the manuscript is not enough to derive these conclusions.

5. Most of the conclusions derived from the data presented in figure 3 are based on the statistical analysis that seems to be very biased because of multiple testing. I recommend that the authors correct for possible false discovery rate.
6. In figure 1a, figure 1b, figure 3a and figure 3b, the depicted sugar structures and the labels are not clear to the reader.
7. In figure 5, which is supposed to describe the effect of the Lipid A variant on Amoeba's consumption of E coli does not match the figure that is submitted. It seems that Figure 5 and Figure 6 are the same.
8. Minor comment: A description of what kind of E coli strain ATCC11775 is will help readers.
9. Since the deletions cause a direct effect on the cell wall structure, the effects which are shown could be a direct result of the sensitivity of the mutant strains to stress and not specifically to resistance/sensitivity to predation. Therefore, the authors should ideally use a different stress than the predator used in this study to see whether the sensitivity of the strains specifically to the predator or not.
10. Line 64: I am not convinced that the references listed demonstrate that bacteria have evolved mechanisms for the purpose of avoiding predation. Therefore, I recommend that the authors avoid such statements.

Staff Comments:

Preparing Revision Guidelines

Please return the manuscript within 60 days; if you cannot complete the modification within this time period, please contact me. If you do not wish to modify the manuscript and prefer to submit it to another journal, please notify me of your decision immediately so that the manuscript may be formally withdrawn from consideration by Microbiology Spectrum.

Reviewer #1 (Comments for the Author):

Figure 1A and B.

This Figure is very important to follow the paper, but it is also difficult to read due to its small labeling. Please amend. Moreover, I suggest to include labels for Kdo and HexI-II to 1A, since these terms are used repeatedly.

Changes were made to increase clarity of labelling of the figures.

Fig. 2 and all similar graphs:

In what unit is the relative area measured, on the cm or mm scale? I would find this information helpful to interpret the data

The relative area cited in the original figures is a ratio of plaque sizes of test vs control and is thus unitless. However to facilitate understanding and statistical analysis, all figures now depict actual area as measured in mm²

Fig. 3A/B: Again, these Figures are not readable for small labels, and 3A also for poor resolution. Please amend. Furthermore, it appears that in 3B the labels below (description of mutants) were accidentally overlaid?

Figure has been modified to increase clarity

Fig. 3C: The labeling of statistical significance is unclear. What significance is supposed to be shown by the bar second from top, which is drawn between waaR, waaO and waaG mutants? Please clarify.

All figures have been modified to increase the ease of determining statistical significance.

I. 237/ Fig. 3D: Is the difference between R3 and R3waaI statistically significant? If yes, please include the respective label to the figure. If no, please clearly state in the text.

Yes, the difference between R3 and R3waaI is statistically significant. That information is presented in a modified figure.

Fig. 3D and corresponding text I. 203-206: I agree with the statement that deletion of waaR enhances consumption of K12 and R3. However, I do not understand why in this assay consumption of K12 and R3 are so different from each other, when in Fig. 2 the authors point out that they are almost identical? Please clarify.

The difference between K12 and R3 the reviewer perceived in Figure 3 vs Figure 2 is an unfortunate consequence of the normalization procedures used in creating these figures. We apologize the initial way we reported these results created the mistaken impression that the data results were different between Figs 2 & 3. Regardless, as described above, we created new figures that directly report the measured plaque sizes, thereby facilitating the correct comparisons. As can now be seen in the updated Figures 3C&D, the difference in amoeba consumption, as measured by plaque area is quite similar, mirroring that seen in Figure 2.

I. 254 and throughout: Please be consistent the designation of mutant strains with or without colon.

Nomenclature has been checked and corrected throughout. Our apologies for the confusing inconsistencies.

I. 261: It is not clear to me how this experiment would work - do the authors expect that adding purified LPS components would block a receptor, e.g. aMBP? If so, this should be explained more clearly.

Indeed the referee is correct, that is the basis of this experiment. Our previously published results (see Arnold, et al., (2016) Determinants that govern the recognition and uptake of *Escherichia coli* O157 : H7 by *Acanthamoeba castellanii*. *Cellular Microbiology*, 18: 1459– 1470. doi: [10.1111/cmi.12591](https://doi.org/10.1111/cmi.12591), for details) validated this as an approach. The text has been modified to clarify the concept behind this experiment.

Figure 5 is identical with Figure 6, presumably a mistake in image upload. Please present the correct figure for the data described in the text for Fig. 5.

We apologize for this oversight! The correct figure is now included.

Reviewer #2 (Comments for the Author):

1. It is not clear if the authors have corrected for multiple testing in Figure - 2. To the naked eye, it looks as if the multiple testing will reveal that difference is not significant in most cases.

In response to this reviewer's comments, we completely reanalyzed and corrected significance tests on all data in all figures. As seen in the newly revised figures, this reanalysis validated the significance of the original conclusions. Regardless, we now report significance using p values corrected for multiple testing

Moreover, it is not clear why did the authors use MG 1655 as a reference for their analysis? Especially when ATCC11775 can be used as a control, which is in fact even more resistant to predation than the MG strain.

To make the results easier to compare, we have eliminated all normalizations and instead now report direct measurements of plaque sizes. Moreover, we have removed the results from ATCC11775 as this strain has an O-antigen and since this manuscript focuses exclusively on the role of sugars of the OS region in modulating recognition of bacteria by amoebae, they are 'wide of the point' for this paper.

2. Similarly data from the MG1655 strain is used to normalize the data in figure 3. Why not use respective parental strain as a reference? For instance, when the deletion mutants of R3 were described, MG1655 was taken as a control to normalize the data. Why not use R3 instead to normalise the data?

As described in the response to comment 2, we have eliminated all normalizations and instead report direct measurements of plaque sizes.

3. A minor comment about the Figures 3 C and D: "Bacterial strains" is mentioned twice. Authors can remove the heading from this figure. This will allow them to show the results from the statistical test clearly. For example, it is not clear whether the difference between $\Delta waaR$, $\Delta waaG$, and $\Delta waaO$ in figure 2C is significant or not.

We have modified the figure to correct this error as well as to better depict the statistical significance of the differences between the strains

4. Line 206 to 208 is not clear. Seems like a part of the sentence is missing.

Text has been modified to increase clarity.

More importantly, it is not clear to me that the differences seen here are because of differences in the ability of predators in recognizing the prey. I suspect that such differences may arise because of differential ability to digest the prey, differential nutritional quality, or differences in the ability to move on the respective prey types. The information provided in the manuscript is not enough to derive these conclusions.

We are confused as to why the reviewer specifically raises this objection in the context of the results obtained with $waa(X)$ deletion strains. A similar argument could be made for concerning the results obtained with the various R-type strains. Regardless, given that the $waa(X)$ deletions only remove one or more sugars from a given OS framework, it is difficult to imagine that these changes could affect the amoebae's ability to digest these strains or significantly affect its nutritional quality. Regardless, we now acknowledge these possibilities in the discussion. However, we are unclear by what the reviewer means by "differences in the ability to move on the respective prey types" and therefore we did not specifically address this concern in the revised manuscript.

5. Most of the conclusions derived from the data presented in figure 3 are based on the statistical analysis that seems to be very biased because of multiple testing. I recommend that the authors correct for possible false discovery rate.

As described in the response to Comment 1, we completely reanalyzed and corrected significance tests on all data in all figures. As seen in the newly revised figures, this reanalysis validated the significance of the original conclusions. Regardless, we now report significance using p values corrected for multiple testing

6. In figure 1a, figure 1b, figure 3a and figure 3b, the depicted sugar structures and the labels are not clear to the reader.

Changes were made to increase clarity of labelling of the figures.

7. In figure 5, which is supposed to describe the effect of the Lipid A variant on Amoeba's consumption of E coli does not match the figure that is submitted. It seems that Figure 5 and Figure 6 are the same.

We apologize for this oversight! The correct figure is now included.

8. Minor comment: A description of what kind of E coli strain ATCC11775 is will help readers.

As described in the response to the second part of Comment 1, since this manuscript focuses exclusively on the role of sugars of the OS region in modulating recognition of bacteria by amoebae, and ATCC11775 as this strain has an O-antigen we have removed the results obtained with this strain as they are 'wide of the point' for this paper.

9. Since the deletions cause a direct effect on the cell wall structure, the effects which are shown could be a direct result of the sensitivity of the mutant strains to stress and not specifically to resistance/sensitivity to predation. Therefore, the authors should ideally use a different stress than the predator used in this study to see whether the sensitivity of the strains specifically to the predator or not.

We do not see how a change in cell wall structure that affect stress could or would impact amoeba consumption. Moreover, these strains all grow similarly and to essentially identical densities to each other and their progenitors and are unaffected by changes in osmolarity. Hence there is no evidence that these strains are differentially affected by cell wall stress.

10. Line 64: I am not convinced that the references listed demonstrate that bacteria have evolved mechanisms for the purpose of avoiding predation. Therefore, I recommend that the authors avoid such statements.

The concept that many bacterial traits that in humans function pathogenicity/virulence factors developed before the rise of mammals to help bacteria evade predation is both well accepted and well-supported in the literature (see Hoque. et al. Adaptation to an amoeba host drives selection of virulence-associated traits in *Vibrio cholerae*. ISME J 16, (2022) 856–867. <https://doi.org/10.1038/s41396-021-01134-2> for a recent example). The data presented in this manuscript presents evidence that cell surface traits are part of the bacterial antipredator defense repertoire.

December 21, 2022

Dr. Gerald B Koudelka
University at Buffalo
Biological Sciences
611 Cooke Hall, North Campus
Buffalo, NY 14260

Re: Spectrum02930-22R1 (The oligosaccharide region of LPS governs predation of *E. coli* by the bacterivorous protist, *Acanthamoeba castellanii*)

Dear Dr. Gerald B Koudelka:

Again, I am sorry for the extra time I took managing this manuscript. I have now analyzed your answers to the reviewers and the new version on the manuscript. I believe you have successfully addressed the issues raised, and I am happy to communicate that your manuscript has been accepted. I am forwarding it to the ASM Journals Department for publication. You will be notified when your proofs are ready to be viewed.

Thank you for submitting your paper to Spectrum.
Have a nice holiday.

Sincerely,

Olaya Rendueles
Editor, Microbiology Spectrum
